# A systematic review of alternative surveillance approaches for lymphatic filariasis in low prevalence settings: Implications for post-validation settings

**Nicholas Riches**[1]*, **Xavier Badia-Rius**[1], **Themba Mzilahowa**[2], **Louise A. Kelly-Hope**[1]

**1** Centre for Neglected Tropical Diseases, Department of Tropical Disease Biology, Liverpool School of Tropical Medicine, Liverpool, United Kingdom, **2** Malaria Alert Centre, College of Medicine, Blantyre, Malawi

* nicholas.riches@lstmed.ac.uk

## Abstract

Due to the success of the Global Programme to Eliminate Lymphatic Filariasis (GPELF) many countries have either eliminated the disease as a public health problem or are scheduled to achieve this elimination status in the coming years. The World Health Organization (WHO) recommend that the Transmission Assessment Survey (TAS) is used routinely for post-mass drug administration (MDA) surveillance but it is considered to lack sensitivity in low prevalence settings and not be suitable for post-validation surveillance. Currently there is limited evidence to support programme managers on the design of appropriate alternative strategies to TAS that can be used for post-validation surveillance, as recommended by the WHO. We searched for human and mosquito LF surveillance studies conducted between January 2000 and December 2018 in countries which had either completed MDA or had been validated as having eliminated LF. Article screening and selection were independently conducted. 44 papers met the eligibility criteria, summarising evidence from 22 countries and comprising 83 methodologically distinct surveillance studies. No standardised approach was reported. The most common study type was community-based human testing (n = 42, 47.2%), followed by mosquito xenomonitoring (n = 23, 25.8%) and alternative (non-TAS) forms of school-based human testing (n = 19, 21.3%). Most studies were cross-sectional (n = 61, 73.5%) and used non-random sampling methods. 11 different human diagnostic tests were described. Results suggest that sensitivity of LF surveillance can be increased by incorporating newer human diagnostic tests (including antibody tests) and the use of mosquito xenomonitoring may be able to help identify and target areas of active transmission. Alternative sampling methods including the addition of adults to routine surveillance methods and consideration of community-based sampling could also increase sensitivity. The evidence base to support post-validation surveillance remains limited. Further research is needed on the diagnostic performance and cost-effectiveness of new diagnostic tests and methodologies to guide policy decisions and must be conducted in a range of countries. Evidence on how to integrate surveillance within other routine healthcare processes is also important to support the ongoing sustainability of LF surveillance.

**Data Availability Statement:** All relevant data are within the manuscript and its Supporting Information files.

**Funding:** The author(s) received no specific funding for this work.

**Competing interests:** The authors have declared that no competing interests exist.

## Author summary

Lymphatic filariasis (LF) is a mosquito-borne disease, which can result in complications including swelling affecting the limbs (lymphoedema) or scrotum (hydrocele). LF can be eliminated by mass drug administration (MDA) which involves whole communities taking drug treatment at regular intervals. After MDA programmes, country programmes conduct the Transmission Assessment Survey (TAS), which tests school children for LF. It is important to continue testing for LF after elimination because there can be a 10-year period between becoming infected and developing symptoms, but it is thought that the use of TAS in such settings is likely to be too expensive and also not sensitive enough to detect low-level infections. Our study assesses the results from 44 studies in areas of low LF prevalence that have investigated methods of surveillance for LF which differ from the standardised TAS approach. These include both human and mosquito studies. Results show that there is currently no standardised approach to testing, but that surveillance can be made more sensitive through the use of new diagnostic tests, such as antibody testing, and also by targeting higher risk populations. However, further research is needed to understand whether these approaches work in a range of settings and whether they are affordable on the ground.

## Introduction

Lymphatic filariasis (LF) is a mosquito-borne parasitic infection which is caused by three species of filarial worms: *Wuchereria bancrofti*, *Brugia malayi* and *Brugia timori*[1, 2]. It can damage the human lymphatic system, resulting in disabling complications including lymphoedema and hydrocele[1]. An estimated 886 million people live in areas at risk of LF infection and 36 million people are currently suffering from LF-related complications[2].

The Global Programme to Eliminate LF (GPELF) was established in 2000 with the intention of eliminating LF as a public health problem[3]. This has involved actions to interrupt transmission, through the systematic delivery of mass drug administration (MDA) at a population level, and to ensure that cases of morbidity linked to LF receive appropriate treatment[4].

Since 2010, demonstrating interruption of transmission has required three successful Transmission Assessment Surveys (TAS). These are school-based surveys which use rapid antigen tests (e.g. BinaxNOW) to sample a population of 6-7-year-old children at least 6 months after the final MDA[4, 5]. Successful delivery of these TASs allows a country to be validated as having eliminated LF as a public health problem.

By the end of 2018, 14 countries had been validated as having eliminated LF, with a further 59 requiring ongoing interventions and surveillance[2]. In the coming decade, many of these countries are expected to be validated as having achieved elimination status. This work is supported by the continued funding commitment from international donors and new drug regimens such as triple therapy which could be scaled up in challenging areas, including India which has the largest burden of disease[1, 6].

Following validation of elimination of LF as a public health problem, the WHO recommend that countries continue surveillance for LF to detect any possible recrudescence of infection but there are no clear recommendations on specific surveillance methods and thresholds to be used[4, 6]. It is acknowledged that the TAS methodology is resource-intensive and may also lack sensitivity in low-prevalence settings[5, 7]. Consequently, there is increasing interest in the appropriateness and effectiveness of alternative methods of LF surveillance, and whether these can be integrated within health systems in post-validation settings.

This review focuses on alternative (non-TAS) LF surveillance studies conducted in low-prevalence settings since 2000, including both human and mosquito studies. This cut-off represents the establishment of GPELF and the introduction of a more standardised approach to LF surveillance and the emergence of newer diagnostic tests. It aims to describe these studies in relation to factors including diagnostic tests, sampling methods and reported results, and to compare results with concurrent TAS outcomes where possible, in order to make recommendations to programme managers and highlight areas requiring further research.

## Methods

### Protocol and registration

This review was conducted and reported according to Preferred Reporting Items for Systematic Reviews and Meta-analyses Statement (PRISMA) guidelines (S1 File).

### Search strategy

The following databases were searched for papers published from 2000 to November 2018: PubMed, Scopus and the Cochrane Database of Systematic Reviews. A combination of MeSH terms and text words were used to describe concepts relating to both LF and surveillance (S2 File). Any additional papers found to be relevant during this process were included.

### Inclusion criteria

Studies were included in the systematic review if they (1) were a primary research study investigating methods of population-based LF surveillance other than routine TAS surveys; (2) included surveillance methods pertaining to either humans (reservoir) and/or mosquitoes (vector); and (3) were conducted in a low prevalence setting, either post-MDA or post-validation. The review was limited to English-language publications with full-text availability conducted after 2000, following the establishment of the GPELF. Studies describing diagnostic test studies were not included if their design did not include population-level sampling.

### Study selection and data extraction

A two-stage process was followed for data selection. Firstly, titles and abstracts of all eligible studies were independently reviewed (co-authors NR and XBR). Any article deemed 'potentially' relevant then underwent independent full-text review (NR and XBR). Discrepant ratings for any papers at stage two were discussed until consensus was reached. A standardised data extraction form was developed, piloted and refined. Where papers reported on more than one study design, these were extracted separately. NR extracted from all the papers and XBR extracted from a sample of 10% of the total. No significant discrepancies were identified during this process.

Extraction focused on the core themes identified during scoping work: (1) location (WHO Region and country, predominant mosquito type); (2) programme context (number of MDA rounds, date of last MDA and elimination status; (3) study design; (4) sampling strategy (including sample size and sampling methods); (5) diagnostic tests used; (6) outcomes of surveillance activity, including comparison with TAS results where applicable; and (7) integration of surveillance with other disease programmes.

### Risk of bias assessment

Risk of bias was assessed using a modified version of the Crowe and GATE validated appraisal tools. Scores of 0–2 were assigned for all studies based on study design (not stated, cross-sectional, longitudinal). Human sampling studies were further assessed in relation to sample size

terciles (0-760/761-2,464/>2,464), method of sampling participants (not stated/non-random/ random) and study population (not stated/children or adults/children and adults). Mosquito sampling studies also assigned scores according to sample size terciles (0–4,679/4,680–10,871/ >10,871), catch-site sampling (not stated, non-random, random) and method of analysis (not stated/dissection/PCR analysis). It was decided not to include location sampling in the assessment since it may be preferable to use non-random methods in some scenarios (e.g. conducting surveillance activities in response to a suspected hotspot). Total risk of bias scores (marked out of 8) were calculated for each study and are presented in Tables 2 and 4. A full breakdown of scores for each study is listed in S1 Table.

## Data synthesis and analysis

Details of publication details, programme context and study design are presented for all studies combined. This is followed by data on sampling strategy, diagnostic test usage and outcomes, split for human and mosquito surveillance studies separately. The impact of age and gender on diagnostic test performance in humans is explored. Analysis then included: (1) comparison between human and mosquito surveillance studies; (2) comparison with TAS results, where applicable; and (3) evidence of integration of surveillance methods within health systems. The analysis aims to determine factors which can increase the sensitivity (defined as the proportion of true positive cases identified by a diagnostic test) in low prevalence settings.

# Results

## Selected studies

Fig 1 highlights the PRISMA steps of identification, screening, eligibility and inclusion of papers. A total of 1,378 papers were identified from the initial search, once duplicates had been

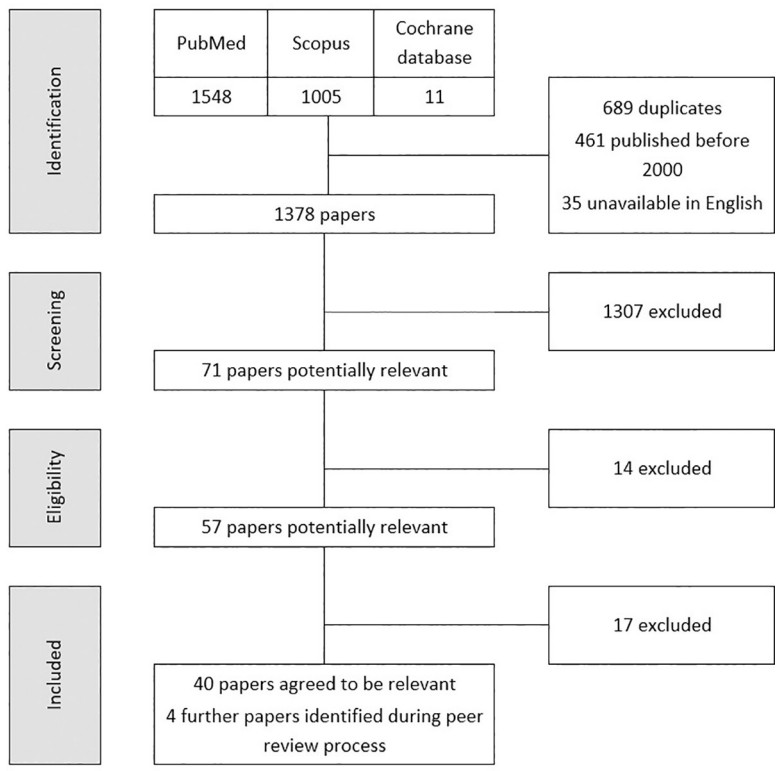

**Fig 1. PRISMA flow diagram.**

excluded. Of these, 71 were considered potentially relevant following title/abstract screening by two independent reviewers. 57 of these were labelled potentially relevant by at least one reviewer following full-text screening. When discrepant results were reviewed, this total reduced to 40 papers which then proceeded to data extraction. An additional four papers were identified during the peer review process. The 44 papers which met eligibility criteria comprised of 83 methodologically distinct study designs (Table 1). These studies are henceforth considered separately except for one paper which pooled results of school and community surveys.

A significant degree of heterogeneity was identified in the included studies. This included variation in study design, baseline endemicity, population sampled, use of diagnostic tests and reporting metrics. It was agreed that this variation precluded formal meta-analysis and instead required a narrative review structured according to the core themes identified.

**Publication details.** 26 papers (59.1%) were published between 2015–2019, 15 (34.1%) were published between 2010–2014 and 3 (6.8%) between 2005–2009. 23 (52.3%) papers reported on human surveillance only, 9 (20.5%) on mosquito surveillance only and 12 (27.2%) reported on both human and mosquito surveillance together.

**Location (WHO Region and country).** Papers reported data from 22 countries in total; 21 (41.1%) came from the Western Pacific Region, 13 (25.5%) from the African Region, 12 (23.5%) from the South East Asian Region, 4 (7.8%) from the Eastern Mediterranean Region and 1 (2.0%) from the Region of the Americas. Fig 2 shows the geographical distribution of countries included in the review.

**Programme context.** 72 (86.7%) studies reported data from countries which had completed MDA but had not yet completed TAS. 11 studies (13.3%) were from countries validated as having eliminated LF, of which two studies described surveillance following successful completion of TAS. 36 (70.6%) studies reported on previous MDA activity, for which the median number of MDA rounds prior to surveillance was 5 (range 3 to 13).

**Predominant mosquito type.** Studies were conducted in areas with a range of different mosquito vector genera, most commonly *Anopheles* sp (n = 15, 29.4%) from the African and

**Table 1. Characteristics of included studies.**

| Description | No. of studies (%) |
| --- | --- |
| **Study start date** | |
| 2000–2004 | 6 (7.2%) |
| 2005–2009 | 31 (37.3%) |
| 2010–2014 | 19 (22.9%) |
| 2015–2019 | 16 (19.3%) |
| Not stated | 11 (13.3%) |
| **Study type** | |
| Cross-sectional | 61 (73.5%) |
| Longitudinal | 22 (26.5%) |
| **Surveillance method** | |
| Community survey | 42 (47.2%) |
| School survey | 19 (21.3%) |
| Laboratory surveillance | 2 (2.2%) |
| Health centre surveillance | 1 (1.1%) |
| Active surveillance | 1 (1.1%) |
| Occupational surveillance | 1 (1.1%) |
| Xenomonitoring survey | 23 (25.8%) |

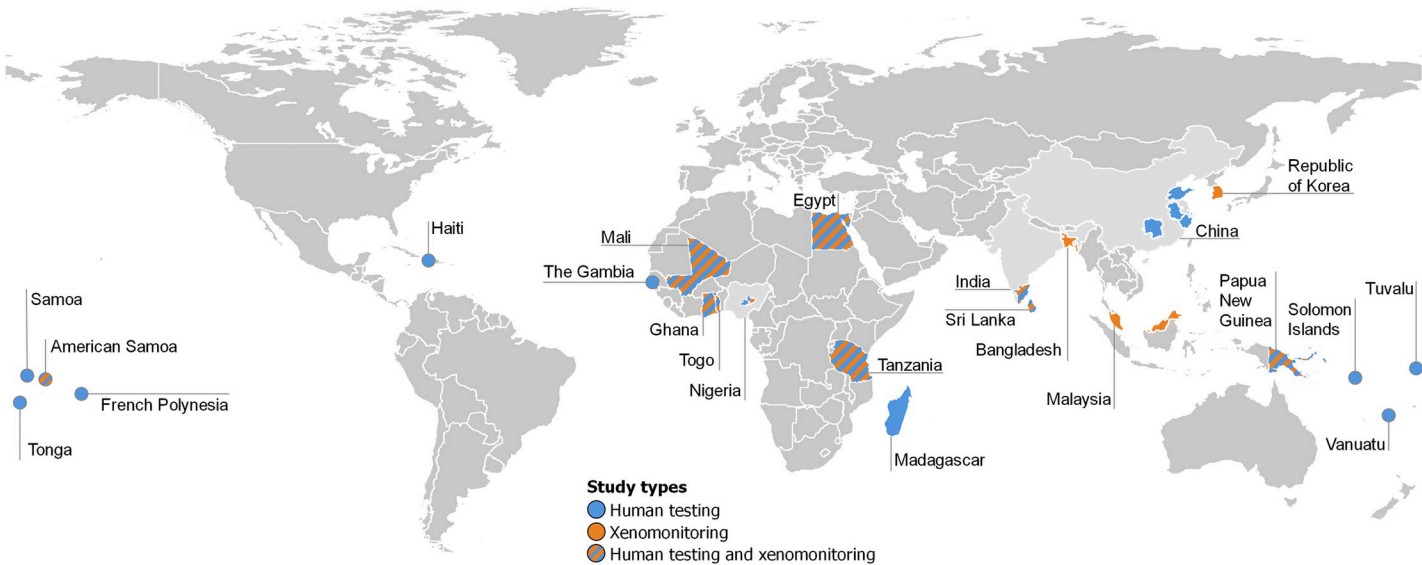

**Fig 2. Map of countries reporting data.** For highly populated countries (e.g. Nigeria and India) where mapping was not nationally representative, the specific area/ region being sampled is highlighted. These data were extracted from the Geoconnect website (https://www.geoconnect.org/).

Western Pacific Regions, *Culex* sp (n = 13, 25.5%) from studies in the South East Asian and Eastern Mediterranean Regions, and *Aedes* sp (n = 12, 23.5%) from the Western Pacific Region. A further 12 studies (23.5%) were conducted in settings with more than one vector for LF according to the WHO Practical Entomology manual[8].

**Study design.**   61 studies (73.5%) were cross-sectional in design and 22 (26.5%) were longitudinal studies. The most common study designs were community surveys (n = 42; 47.2%), xenomonitoring surveys (n = 23; 25.8%) and school surveys (n = 13; 14.6%).

### Human surveillance studies

Table 2 summarises the characteristics of the 35 papers which reported data on human surveillance for lymphatic filariasis, comprising 60 distinct studies. Full results from these studies can be found in S2 Table.

**Sample size.**   The median sample size was 1,472 (range = 40 to 35,582; interquartile range = 596–3,207). The majority of studies (n = 36; 60.0%) included both children and adults in their study design. 15 studies (25.0%) focused on children only and five (8.3%) on adults only. In total, the studies reported data on 208,568 participants.

**Sampling methods.**   Where stated (n = 42), the most common approach to selecting a sampling location involved non-random methods, such as purposive or convenience sampling (n = 30, 71.4%). In most cases surveillance was conducted in response to identification of a hotspot of infection. Other methods involved using random sampling methods (n = 7, 17.5%) while four studies described national surveillance studies [10, 11, 31]. Participants were then sampled using either non-random methods (n = 34, 69.3%) or random methods (n = 15, 30.6%).

**Diagnostic tests.**   Included studies described results using 12 different diagnostic tests. 58 studies involved blood samples of which the majority were finger prick samples. The most common tests were microscopy for microfilaraemia (MF) (n = 38; 63.3%); BinaxNOW (n = 36; 60.0%); Bm14 Ab (n = 20; 33.3%), Og4C3 Ag (n = 17; 28.3%), Wb123 Ab (n = 9; 15.0%) and Wb PCR (8, 13.3%). Table 3 compares results where the same diagnostic test was

**Table 2. Human surveillance study characteristics.**

| Country (last MDA)[1] | Reference (Quality score) | Study date | Context | Study design | Age criteria | Total sample size | Tests performed |
|---|---|---|---|---|---|---|---|
| **American Samoa (2007)** | Mladonicky et al. 2009 [9] (4/8) | 2006 | Post-MDA | Cross-sectional community survey | ≥5 years | 579 | BinaxNOW, MF, Bm14 Ab |
| | Coutts et al. 2017 [10] (6/8) | 2007 | Post-MDA | Cross-sectional community survey | ≥2 years | 1,881 | BinaxNOW |
| | Lau et al. 2014 [11] (5/8) | 2010 | Post-MDA | Cross-sectional community survey | ≥18 years | 807 | Og4cC3 Ag>128 units, Og4cC3 Ag>32 units, Wb123 Ab, Bm14 Ab |
| | Lau et al. 2017a [12] (4/8) | 2014 | Post-MDA | Cross-sectional occupational survey | ≥15 years | 602 | BinaxNOW, Og4C3 Ag, Bm14 Ab, Wb123 Ab |
| | Lau et al. 2017b [12] (4/8) | 2014 | Post-MDA | Cross-sectional community survey | ≥2 years | 476 | BinaxNOW, Og4C3 Ag, Bm14 Ab, Wb123 Ab |
| | Lau et al. 2017c [12] (4/8) | 2014 | Post-MDA | Cross-sectional school survey | 7–13 years | 283 | BinaxNOW |
| | Won et al. 2018 [13] (5/8) | 2015 | Post-MDA | Longitudinal school survey | 5–10 years | 1,134(TAS 1) 864(TAS 2) | BinaxNOW, Wb123 Ab, Bm14 Ab, Bm33 Ab |
| | Sheel et al. 2018 [14] (7/8) | 2016 | Post-MDA | Cross-sectional community survey | ≥8 years | 2,507 | MF, FTS (filarial test strips) |
| **China** | Huang et al. 2016a [15] (2/8) | 2002 | Post-validation | Cross-sectional school survey | Children | 542 | Chinese filariasis IgG4 ELISA kit, MF |
| | Huang et al. 2016b [15] (1/8) | 2003 | Post-validation | Cross-sectional community survey | Not stated | 436 | Chinese filariasis IgG4 ELISA kit |
| | Huang et al. 2016c [15] (3/8) | 2004 | Post-validation | Cross-sectional community survey | Not stated | 5,787 | Chinese filariasis IgG4 ELISA kit |
| | Huang et al. 2016d [15] (4/8) | 2002 and 2004 | Post-validation | Cross-sectional community survey | Children and adults | 762 | Chinese filariasis IgG4 ELISA kit, MF |
| | Huang et al. 2016e [15] (2/8) | 2002–2008 | Post-validation | Longitudinal community survey | Not stated | 218 | Chinese filariasis IgG4 ELISA kit |
| | Itoh et al. 2007 [16] (4/8) | 2004 | Post-validation | Cross-sectional school survey | 6 to 10 years (Yongjia) 5–15 years (Gaoan) | 2,411 (Yongjia) 7,998 (Gaoan) | IgG4 ELISA (urinary) |
| **Egypt (2005)** | Moustafa et al. 2014a [17] (4/8) | 2012 | Post-MDA | Cross-sectional school survey | 6–7 years | 1,321 | BinaxNOW, Bm14Ab |
| | Moustafa et al. 2014b [17] (3/8) | 2012 | Post-MDA | Cross-sectional community survey | 16–60 years | 75 | BinaxNOW |
| | Ramzy et al. 2006a [18] (7/8) | Not stated | Post-MDA | Longitudinal community survey | ≥4 years | 1,064 (Giza) 744 (Qalubiya) | BinaxNOW, MF |
| | Ramzy et al. 2006b [18] (4/8) | Not stated | Post-MDA | Longitudinal school survey | 7 and 11 years | 1,653 | BinaxNOW, Bm14 Ab |
| **French Polynesia** | Gass et al. 2011a [19] (5/8) | 2007–2008 | Post-MDA | Cross-sectional school and community survey | 3–80 years | 1,383 | Bm14 Ab, PanLF, Urine SXP, ICT, Og4C3 Ag, MF, PCR |
| **Gambia** | Won et al. 2018 [20] (6/8) | 2015 | Post-validation | Cross-sectional community survey | ≥1 year | 2,612 | Wb 123 Ab ELISA, Bm14 Ab ELISA |
| **Ghana** | Gass et al. 2011b [19] (5/8) | 2007–2008 | Post-MDA | Cross-sectional school and community survey | 3–80 years | 1,466 | Bm14 Ab, ICT, Og4C3 Ag, MF, PCR |
| | Owusu et al. 2015a [21] (4/8) | 2008 | Post-MDA | Cross-sectional school survey | 6–7 and 10–11 years | 308 | BinaxNOW, Og4C3 Ag, Bm14 Ab, Wb123 Ab |
| | Owusu et al. 2015b [21] (5/8) | 2008 | Post-MDA | Cross-sectional community survey | 3–80 years | 653 | BinaxNOW, MF, Og4C3 Ag, Bm14 Ab, Wb123 Ab |
| **Haiti** | Gass et al. 2011c [19] (5/8) | 2007–2008 | Post-MDA | Cross-sectional survey | 3–80 years | 1,322 | Bm14 Ab, PanLF, Urine SXP, ICT, Og4C3 Ag, MF, PCR |

*(Continued)*

**Table 2.** (*Continued*)

| Country (last MDA)[1] | Reference (Quality score) | Study date | Context | Study design | Age criteria | Total sample size | Tests performed |
|---|---|---|---|---|---|---|---|
| **India** **(2011)** **(2007)** **(2004)** | Ramaiah et al. 2013 [22] (6/8) | 2005–2008 | Post-MDA | Longitudinal | Adults and children | Approx. 700 | MF, BinaxNOW |
| | Swaminathan et al. 2012 [23] (6/8) | 2015–2017 | Post-MDA | Cross-sectional community survey | ≥2 years | 35,582 | MF, Og4C3 Ag |
| | Mehta et al. 2018 [24] (3/8) | Study year not reported | Post-MDA | Cross-sectional community survey | ≥5 years | 290 | BinaxNOW, MF |
| **Madagascar** **(2016)** | Garchitorena et al. 2018 [25] (5/8) | 2016 | Post-MDA | Cross-sectional community survey | ≥5 years | 545 | FTS |
| **Mali** **(2008)** | Coulibaly et al. 2015 [26] (5/8) | 2007 | Post-MDA | Longitudinal community survey | ≥2 years | 760 | BinaxNOW, MF |
| | Coulibaly et al. 2016a [27] (6/8) | 2009–2013 | Post-MDA | Longitudinal community survey | 6–7 years | 3,457 | BinaxNOW, MF (if BinaxNOW positive), Wb PCR, Wb123 Ab, Og4C3 Ag |
| | Coulibaly et al. 2016b [27] (5/8) | 2009–2013 | Post-MDA | Longitudinal community survey | ≥8 years | 1,184 | BinaxNOW, MF (if BinaxNOW positive), Wb PCR, Wb123 Ab, Og4C3 Ag |
| **Nigeria** **(2009)** | Richards et al. 2011 [28] (6/8) | 2009 | Post-MDA | Longitudinal community survey | ≥2 years | 1,720 | BinaxNOW, MF |
| **Papua New Guinea** | Mitja et al. 2011 [29] (6/8) | 2011 | Post-MDA | Longitudinal community survey | Not stated | 6,263 | BinaxNOW |
| **Samoa** **(2008)** | Joseph et al. 2011A [30][2] (7/8) | 2007 | Post-MDA | Cross-sectional community survey | Any age | 6,648 | BinaxNOW, MF (if BinaxNOW +ve), BM14 Ab (children aged 5–10 years only) |
| | Joseph et al. 2011Ba [31][2] (7/8) | 2008 | Post-MDA | Cross-sectional community survey | ≥2 years | 2,474 | BinaxNOW, MF, BM14 Ab |
| **Solomon Islands (N/A)** | Harrington et al. 2013 [32] (4/8) | 2011 | Post-validation | Cross-sectional community survey | Adults and children | 307 | Og4C3Ag, MF (if ICT positive/ borderline plus 10% of negative screens) |
| **Sri Lanka** **(2015)** | Rao et al. 2016 [33] (7/8) | 2013 | Post-MDA | Cross-sectional community survey | 2–70 years | 12,977 | MF |
| | Gass et al. 2011d [19] (5/8) | 2007–2008 | Post-MDA | Cross-sectional school and community survey | 3–80 years | 1,477 | PanLF, ICT, Og4C3 Ag, MF, PCR |
| | Chandrasena et al. 2016a [34] (6/8) | 2009–2015 | Post-MDA | Longitudinal community survey | 4–80 years | 2,461 | MF |
| | Chandrasena et al. 2016b [34] (2/8) | 2015 | Post-MDA | Cross-sectional community survey | 7–12 years | 250 | Brugia Rapid |
| | Rahman et al. 2018a [35] (4/8) | Not stated | Post-TAS | Cross-sectional community survey | 5–84 years | 630 | MF, FTS |
| | Rahman et al. 2018b [35] (4/8) | Not stated | Post-TAS | Cross-sectional school survey | 5–13 years | 2,301 | IgG4 ELISA (urinary) |
| | Rao et al. 2014a [36] (7/8) | 2011–2013 | Post-MDA | Cross-sectional community survey | ≥10 years | 7,156 | BinaxNOW, MF |
| | Rao et al. 2014b [36] (5/8) | Not stated | Post-MDA | Cross-sectional school survey | Grade 1 and 2 | 17,000 | BinaxNOW, BM14 Ab |
| | Rao et al. 2017a [37] (4/8) | 2015–2017 | Post-MDA | Cross-sectional school survey | 6–8 years | 2,227 | BinaxNOW, MF if BinaxNOW +ve, BM14 Ab |
| | Rao et al. 2017b [37] (7/8) | 2015–2017 | Post-MDA | Cross-sectional community survey | ≥10 years | 3,123 | BinaxNOW, MF if BinaxNOW +ve |
| | Rao et al. 2018a [38] (3/8) | 2015 | Post-MDA | Cross-sectional school survey | First and second grade children | 401 | BinaxNOW, BM14 Ab, MF |
| | Rao et al. 2018b [38] (5/8) | 2015 | Post-MDA | Cross-sectional community survey | 10–70 years | 528 | BinaxNOW, MF |
| | Rao et al. 2018c [38] (7/8) | 2015 | Post-MDA | Cross-sectional community survey | ≥2 years | 16,927 | MF |

(*Continued*)

**Table 2.** (Continued)

| Country (last MDA)[1] | Reference (Quality score) | Study date | Context | Study design | Age criteria | Total sample size | Tests performed |
|---|---|---|---|---|---|---|---|
| **Tanzania (2014)** | Gass et al. 2011e [19] (5/8) | 2007–2008 | Post-MDA | Cross-sectional school and community survey | 3–80 years | 1,384 | Urine SXP, ICT, Og4C3 Ag, PCR |
| | Jones et al. 2018 [29, 39] (6/8) | 2015 | Post-MDA | Cross-sectional community survey | 10–79 years | 854 | BinaxNOW |
| **Togo (2009)** | Budge et al. 2014a [40] (6/8) | 2006–2007 | Post-MDA | Longitudinal laboratory surveillance study | Adults | 6,509 | MF |
| | Budge et al. 2014b [40] (5/8) | 2006–2007 | Post-MDA | Cross-sectional community survey | Adults | 7,800 | BinaxNOW |
| | Budge et al. 2014c [40] (6/8) | 2010–2011 | Post-MDA | Longitudinal health facility surveillance study | Adults | 2,880 | Og4C3 Ag, MF (if Ag +ve) |
| | Mathieu et al. 2011 [41] (5/8) | 2006–2007 | Post-MDA | Longitudinal laboratory surveillance study | Not stated | 8,050 | MF |
| | Dorkenoo et al. 2018A [42] (4/8) | 2010–2015 | Post-MDA | Cross-sectional active surveillance of positive cases | Children and adults | 40 | MF, Og4c3 Ag, FTS |
| **Tonga (2005)** | Joseph et al. 2011Bb [31] (4/8) | 2007 | Post-MDA | Cross-sectional school survey | 5–6 years | 797 | BinaxNOW, MF (if ICT +ve), BM14 Ab |
| **Tuvalu** | Gass et al. 2011f [19] (5/8) | 2007–2008 | Post-MDA | Cross-sectional school and community survey | 3–80 years | 1,481 | PanLF, Urine SXP, ICT, Og4C3 Ag, MF, PCR |
| **Vanuatu (2005)** | Joseph et al. 2011Bc [31] (5/8) | 2007 | Post-MDA | Cross-sectional school survey | 5–6 years | 3,840 | BinaxNOW, MF (if ICT +ve), BM14 Ab |
| | Allen at al. 2017 [43] (7/8) | 2005–2006 | Post-MDA | Cross-sectional community survey | ≥1 year | 7,657 | BinaxNOW, MF (if ICT +ve |

[1] According to country or region-level, where stated in papers

[2] MDA in Samoa was subsequently re-started, commencing in 2008

used in the same population, allowing direct comparison of prevalence values within each study. Compared to Binax Now or Alere ICT (the most commonly used tests at the time of most of these surveys) as the index test, Table 3 shows that antibody tests produce a higher proportion of positive results. Bm14Ab and Wb123Ab values are, on average, 5.1 and 6.7 times higher respectively than the corresponding BinaxNOW values, based on the median value of this ratio across the selected studies. Og4C3Ag values are similar to BinaxNOW values in studies where both are used (median ratio = 0.95, range 0.2–1.6).

**Impact of age.** Age-specific prevalence was extracted for twelve different LF diagnostic tests from studies which reported data allowing 10-year age bands to be calculated (Fig 3). A similar pattern is seen for each test, with rates generally increasing through childhood and adolescence before stabilising during adulthood and occasionally falling in older age.

**Impact of gender.** Reported prevalence of LF tests are also known to generally be higher among men in comparison to women (Fig 4).

## Mosquito surveillance studies

Table 4 summarises the characteristics of the 23 papers which reported data on mosquito surveillance for LF. Full results from these studies can be found in S3 Table.

**Sample size.** The median number of mosquitoes collected was 7,860 per study (range 115–69,680, interquartile range 4,383–18,865).

**Sampling methods.** Similar to human surveillance studies, location sampling typically used non-random methods, following identification of a hotspot area by other methods. The

**Table 3. Comparison of diagnostic test results when used for human surveillance in LF, using BinaxNOW as the index test.**

| Country | Reference | Diagnostic test prevalence | | | | | | |
|---|---|---|---|---|---|---|---|---|
| | | BinaxNOW (Index test) | Bm14 Ab | | Og4C3 Ag[1] | | Wb123 Ab | |
| | | | Prevalence (population tested) | Ratio cf. index test | Prevalence (population tested) | Ratio cf. index test | Prevalence (population tested) | Ratio cf. index test |
| American Samoa | Lau et al. 2017a [12] | 1.3% (n = 602) | 11.7% (n = 598) | 9.0 | 1.2% (n = 598) | 0.9 | 10.9% (n = 598) | 8.4 |
| | Lau et al. 2017b[2] [12] | 8.2% (n = 151) | 25.2% (n = 150) | 2.4 | 11.2% (n = 150) | 1.4 | 32.5% (n = 150) | 4.0 |
| | Mladonicky et al. 2009[2] [9] | 4.2% (n = 569) | 14.1% (n = 538) | 3.4 | - | - | - | - |
| | Won et al. 2018 [13] | 0.2% (n = 937) | 6.8% (n = 1,112) | 34.0 | - | - | 1.0% (n = 1,112) | 5.0 |
| | Won et al. 2018 [13] | 0.1% (n = 768) | 3.0% (n = 836) | 30.0 | - | - | 3.6% (n = 836) | 36.0 |
| Egypt | Moustafa et al. 2014 [17] | 0.0% (n = 1,321) | 2.2% (n = 1,321) | N/A | - | - | - | - |
| French Polynesia | Gass et al. (2011) [19] | 9.0% (n = 1,359) | 46.0% (n = 1,329) | 5.1 | 6.4% (1,355) | 0.7 | - | - |
| Ghana | Gass et al. (2011) | 6.7% (n = 1,372) | 9.9% (n = 1,159) | 1.5 | 8.9% (n = 1,355) | 1.3 | - | - |
| Ghana | Owusu et al. 2015a [21] | 1.6% (n = 308) | 4.9% (n = 308) | 3.1 | 1.0% (n = 308) | 0.6 | - | - |
| | Owusu et al. 2015a [21] | 7.8% (n = 653) | 12.9% (n = 653) | 1.7 | 12.2% (n = 653) | 1.6 | - | - |
| Haiti | Gass et al. (2011) [19] | 21.2% (n = 1,266) | 53.1% (n = 1,214) | 2.5 | 18.8% (n = 1,179) | 0.9 | - | - |
| Samoa | Joseph et al. 2011[2] [31] | 7.7% (2,026) | 62.7% (n = 2,026) | 8.1 | - | - | - | - |
| Sri Lanka | Gass et al. (2011) [19] | 3.0% (n = 1,449) | - | - | 0.5% (n = 1,432) | 0.2 | - | - |
| | Rao et al. 2017[2] [37] | 0.3% (n = 1,893) | 1.9% (n = 2,126) | 6.3 | - | - | - | - |
| | Rao et al. 2014[2] [36] | 0.2% (n = 2,561) | 10.6% (n = 2,110) | 53 | - | - | - | - |
| | Rao et al. 2014b [36] | 0.05% (n = 6,198) | 2.2% (n = 6,198) | 44 | - | - | - | - |
| | Rao et al. 2018a [38] | 1.2% (n = 401) | 5.7% (n = 387) | 4.75 | - | - | - | - |
| Tanzania | Gass et al. (2011) [19] | 8.1% (n = 1,316) | - | - | 8.2% (n = 1,126) | 1.0 | - | - |
| Tonga | Joseph et al. 2011Bb [31] | 0% (n = 797) | 6.3% (n = 797) | N/A | - | - | - | - |
| Tuvalu | Gass et al. (2011) [19] | 5.0% (n = 1,455) | | - | 4.9% (1,333) | 1.0 | - | - |
| Vanuatu | Joseph et al. 2011Bc [31] | 0% (n = 3,840) | 6.0% (n = 3,840) | N/A | - | - | - | - |

[1] A threshold value of >32 units was selected for Og4C3 Ag when multiple values were presented.

[2] Weighted average of component studies

[3] Standard TAS with the addition of antibody testing

majority of studies then described various methods for taking a random sample of households from which to sample mosquitoes, either indoors or outdoors. The most common mosquito sampling method was the gravid trap (n = 9; 39.1%) followed by various baited traps (n = 6;

**Table 4. Mosquito diagnostic study characteristics**

| Country (last known MDA)[1] | Main vector | Reference (Quality score) | Study date | Context | Study design | Catch method | Sample size | Analysis method |
|---|---|---|---|---|---|---|---|---|
| American Samoa (2007) | *Aedes* spp. | Schmaedick et al. 2014 [44] (6/8) | 2011 | Post-MDA | Cross-sectional survey | BG-Sentinel traps | 21,861 mosquitoes | PCR analysis |
| Bangladesh | *Culex* spp. | Irish et al. 2018 [45] (6/8) | 2016 | Post-MDA | Cross-sectional survey | CDC gravid traps | 5,926 mosquitoes | PCR analysis |
| Egypt (2013) | *Culex* spp. | Ramzy et al. 2006 [18] (7/8) | Not stated | Post-MDA | Longitudinal survey | Aspiration of indoor resting mosquitoes | 8,531 mosquitoes | PCR analysis |
| | | Abdel-Shafi et al. 2016 [46] (5/8) | 2014–15 | Post-MDA | Cross-sectional survey | Light traps | Not stated | PCR analysis |
| | | Moustafa et al. 2017 [47] (4/8) | 2014 | Post-MDA | Cross-sectional survey | Gravid traps | 7,970 mosquitoes | PCR analysis |
| Ghana | Multiple | Owusu et al. 2015a [21] (5/8) | 2008 | Post-MDA | Cross-sectional survey | Pyrethrum knockdown method | 401 mosquitoes | PCR analysis |
| | | Owusu et al. 2015b [21] (5/8) | 2008 | Post-MDA | Cross-sectional survey | Gravid trap | 4,099 mosquitoes | PCR analysis |
| India (2011/ 2007/ 2004) | Multiple | Ramaiah et al. 2013 [22] (4/8) | 2005–2010 | Post-MDA | Longitudinal survey | Aspiration of indoor resting mosquitoes | 10,842 mosquitoes | Dissection |
| | | Subramanaian et al. 2017 [48] (8/8) | 2012 | Post-MDA | Longitudinal survey | CDC gravid traps | 41,294 mosquitoes | PCR analysis |
| | | Mehta et al. 2018 [24] (3/8) | Not stated | Post-MDA | Cross-sectional survey | Gravid trap | 2,429 mosquitoes | Dissection |
| Malaysia | Multiple | Beng et al. 2016 [49] (3/8) | Not stated | Post-MDA | Cross-sectional survey | Bare leg catch and CDC light trap | 4,378 mosquitoes | PCR analysis |
| Mali (2008) | *Anopheles* spp. | Coulibaly et al. 2015 [26] (4/8) | 2007 | Post-MDA | Longitudinal survey | Human landing catch | 4,680 mosquitoes | Dissection |
| | | Coulibaly et al. 2016a [27] (5/8) | 2009–2013 | Post-MDA | Longitudinal survey | Human landing catch | 14,424 mosquitoes | Dissection |
| | | Coulibaly et al. 2016b [27] (6/8) | 2012 | Post-MDA | Longitudinal survey | Pyrethrum spray catch | 115 mosquitoes | PCR analysis |
| Nigeria (2009) | *Anopheles* spp. | Richards et al. 2011 [28] (4/8) | 2009 | Post-MDA | Longitudinal survey | Pyrethrum knockdown method | 4,398 mosquitoes | Dissection |
| Papua New Guinea (1998) | *Anopheles* spp. | Reimer et al. 2013 [50] (5/8) | 2007–2008 | Post-MDA | Longitudinal survey | Human landing catch | 20,345 mosquitoes | PCR analysis |
| South Korea (multiple) | Multiple | Cho et al. 2012 [51] (4/8) | 2009 | Post-validation | Cross-sectional survey | Light trap (Black Hole) | 5,380 mosquitoes | PCR analysis |
| Sri Lanka (2015) | *Culex* spp. | Rao et al. 2014c [36] (7/8) | Not stated | Post-MDA | Cross-sectional survey | Gravid traps | 69,680 mosquitoes | PCR analysis |
| | | Rao et al. 2016 [33] (7/8) | 2013–2014 | Post-MDA | Cross-sectional survey | CDC light trap | 28,717 mosquitoes | PCR analysis |
| | | Rao et al. 2017c [37] (8/8) | 2011–2016 | Post-MDA | Longitudinal survey | CDC gravid traps | 48,301 mosquitoes | PCR analysis |
| | | Rao et al. 2018d [38] (6/8) | 2015–2016 | Post-MDA | Cross-sectional survey | CDC gravid traps | 7,750 mosquitoes | PCR analysis |
| Tanzania (2014) | Multiple | Jones et al. 2018 [29, 39] (4/8) | 2015 | Post-MDA | Cross-sectional survey | CDC gravid traps and CDC light traps | 1,650 mosquitoes | PCR analysis |
| Togo (2009) | *Anopheles* spp. | Dorkenoo et al. 2018B [52] (7/8) | 2015 | Post-MDA | Cross-sectional | Pyrethrum spray catch, Human landing catch and exit trap collection | 10,872 mosquitoes | PCR analysis |

[1] According to country or region-level mentioned in papers

26.1%), human landing collection (n = 5; 21.7%) and pyrethrum space spray catches (n = 4; 17.4%). The variation was partly due to the different species of mosquito being sampled.

| Test | Reference | Age range | | | | | | Trend |
|------|-----------|-----------|------|-------|-------|-------|------|-------|
| | | 0-9 | 10-19 | 20-29 | 30-39 | 40-49 | 50+ | |
| Bm14Ab | Gass et al. (2011) | 31.0% | 35.6% | 36.8% | 46.1% | 46.0% | 50.3% | |
| | Lau et al. (2014) | | | 7.5% | 21.8% | 20.6% | 25.8% | |
| BinaxNOW | Gass et al. (2011) | 5.6% | 9.1% | 10.7% | 11.9% | 11.2% | 11.3% | |
| | Rao et al. (2014) | | 0.2% | 0.6% | 1.1% | 1.2% | 1.3% | |
| FTS | Sheel et al. (2018) | | 0.8% | 2.2% | 5.7% | 6.5% | 7.3% | |
| MF | Gass et al. (2011) | 1.0% | 1.4% | 1.1% | 1.5% | 2.7% | 2.4% | |
| Og4C3Ag | Gass et al. (2011) | 5.1% | 8.0% | 7.9% | 10.7% | 9.3% | 11.2% | |
| | Lau et al. (2014) | | | 4.8% | 4.9% | 1.9% | 2.9% | |
| PCR | Gass et al. (2011) | 1.1% | 0.9% | 0.6% | 1.3% | 2.2% | 1.9% | |
| PanLF | Gass et al. (2011) | 13.4% | 15.7% | 20.0% | 22.4% | 20.5% | 28.7% | |
| Urine SXP | Gass et al. (2011) | 5.9% | 19.3% | 21.0% | 26.9% | 35.0% | 42.8% | |
| Wb123Ab | Lau et al. (2014) | | | 8.2% | 9.2% | 10.0% | 8.6% | |

**Fig 3. Reported prevalence of LF tests according to age range.** Some studies reported decade age bands starting on an even year, e.g. 10–19, rather than 11–20. These data are included in the above table under the adjacent decade age band.

**Diagnostic tests.** Most studies involved PCR analysis of mosquitoes (75.0%) rather than dissection (25.0%).

## Comparison between human and mosquito surveillance results

Table 5 summarises studies which performed both human testing and xenomonitoring in the same geographical area. Overall, there was great variability in survey methods and results which limited comparisons. Interpretation is also limited by the fact that there are currently no recommended species-specific Mosquito Infectivity Rate (MIR) thresholds for LF[8, 53]. A number of studies reported similar results between human testing and xenomonitoring. For example, Rao et al 2018 (38) showed ICT rates of 3% and an MIR of 3%, but a similar pattern was not demonstrated in other Sri Lankan studies. There were also examples where human testing did not detect significant transmission but xenomonitoring did. For example, the study by Ramaiah et al. reported a mosquito infection rate of 4.7% of mosquitoes when a community survey performed concurrently found no evidence of human infection on ICT testing.

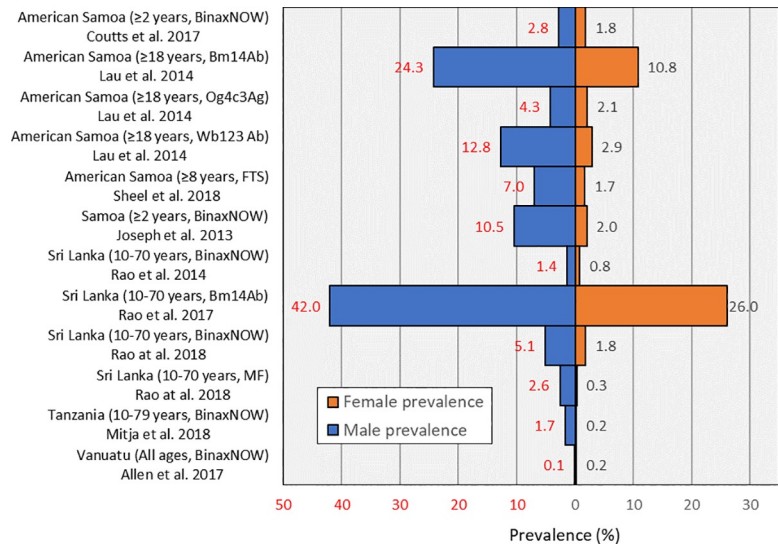

**Fig 4. Reported prevalence of LF tests according to gender.**

**Table 5. Comparison of human and mosquito surveillance study results.**

| Reference (Location) | Human survey type (Age range) | Human sampling results (95% confidence interval) [Sample size] | Xenomonitoring results (95% confidence interval) [Sample size] |
|---|---|---|---|
| Ramzy et al. 2006[18] (Giza, Egypt) | Community survey (≥4 years) | MF = 1.2% (0–2.6%); [n = 1064] | MIR = 0.19% (0.08–0.38%) [n = 4,273] |
| | | BinaxNOW = 4.8% (2.5–7.1%); [n = 1064] | |
| | School survey (7 years) | BinaxNOW = 0.4%; [n=n.s.] | |
| | | Bm14 Ab = 0.2% (0.0–0.5); [n = 896] | |
| | School survey (11 years) | Bm14 Ab = 1.4% (0.3–2.6%); [n = 415] | |
| Ramzy et al. 2006[18] (Qalubyia, Egypt) | Community survey (≥4 years) | BinaxNOW = 3.1% (1.2–4.9%); [n = 764] | MIR = 0% (0.00–0.05%) [n = 4,258] |
| | | MF = 1.2% (0–2.6%); [n = 764] | |
| | School survey (7 years) | BinaxNOW = 0%; [n=n.s.] | |
| | | Bm14 Ab = 0%; [n = 211] | |
| | School survey (11 years) | Bm14 Ab = 0%; [n = 131] | |
| Mehta et al. 2018[24] (Pondicherry, India) | Community survey (≥5 years) | MF = 0.69% (n.s.); [n = 290] | MIR = 0.04% (n.s.) |
| | | ICT = 2.35% (n.s.); [n = 290] | |
| Ramaiah et al. 2013[22] (Muppili, India) | Community survey (15–45 years) | ICT = 0.4% (n.s.); [n = 226] | MIR = 0% (n.s.) [n = 366] |
| Ramaiah et al. 2013[22] (Thenber, India) | Community survey (1–7 years) | ICT = 0% (n.s.); [n = 50] | MIR = 4.7% (n.s.) [n = 339] |
| Ramaiah et al. 2013[22] (Alagramam, India) | Community survey (1–7 years) | ICT = 4.6% (1–7 years); [n = 44] | MIR = 2.2% (n.s.) [n = 361] |
| | Community survey (15–45 years) | ICT = 3.2% (15–45 years); [n = 95] | |
| Coulibaly et al. 2016[27] (Sikasso District, Mali) | Community survey 2009 (6–7 years) | ICT = 0% (0.00–1.64%); [n = 289] | MIR = 0.05% (0.01–0.18%) [n = 4,375] |
| | Community survey 2009 (≥8 years) | ICT = 4.9% (3.53–6.67%); [n = 800] | |
| | Community survey 2011 (6–7 years) | ICT = 2.7% (1.24–5.37); [n = 301] | MIR = 0% (n.s.) [n = 2,803] |
| | Community survey 2011 (≥8 years) | ICT = 3.5% (2.40–5.12%); [n = 795] | |
| | Community survey 2012 (6–7 years) | ICT = 3.9% (2.04–7.00%); [n = 285] | MIR = 0% (n.s.) [n = 5,691] |
| | Community survey 2012 (≥8 years) | ICT = 2.8% (2.08–3.65%); [n = 1,812] | |
| Coulibaly et al. 2015[26] (Sikasso District, Mali) | Community survey (≥2 years) | MF = 0% (n.s.); [n = 760] | MIR = 0.02% (n.s.) [n = 4,680] |
| | | ICT = 7.2% (n.s.); [n = 760] | |
| Richards et al. 2011[28] (Plateau/Nasarawa States, Nigeria) | Community survey (≥2 years) | MF = 0.9% (n.s.); [1,720] | MIR = 0.4% (n.s.) [n = 4,398] |
| | | ICT = 7.4% (n.s.); [1,720] | |
| Mitja et al. 2018[29] (Papua New Guinea) | Community survey (10–79 years) | BinaxNOW = 1.1% (0.6–2.0%) | MIR = 0% |
| Rao et al. 2017[37] (Colombo, Sri Lanka) | School survey (6–8 years) | MF = 0% (0–1.0%); [n = 372] | MIR = 0.34% (0.2–0.6) [n = 4,000] |
| | | ICT = 0% (0–1.0%); [n = 372] | |
| | | Bm14 Ab = 0% (0–1.0%); [n = 360] | |
| | Community survey (≥10 years) | MF = 0% (0–0.7%); [n = 506] | |
| | | ICT = 0% (0–0.7%); [n = 506] | |
| Rao et al. 2017[37] (Gampaha, Sri Lanka) | School survey (6–8 years) | MF = 0% (0–1.0%); [n = 366] | MIR = 0.23% (0.1 - 0.4%) [n = 4,080] |
| | | ICT = 0.3% (0.5–1.5%); [n = 366] | |
| | | Bm14 Ab = 0.6% (0.1–2.1); [n = 335] | |
| | Community survey (≥10 years) | MF = 0% (0–0.7%); [n = 512] | |
| | | ICT = 0.4% (0.1–1.4%) [n = 512] | |

(*Continued*)

**Table 5.** (Continued)

| Reference (Location) | Human survey type (Age range) | Human sampling results (95% confidence interval) [Sample size] | Xenomonitoring results (95% confidence interval) [Sample size] |
|---|---|---|---|
| Rao et al. 2017[37] (Kalutara, Sri Lanka) | School survey (6–8 years) | MF = 0% (0–1.0%); [n = 380] | MIR = 0.26% (0.1 - 0.4%) [n = 3,986] |
| | | ICT = 0% (0–1.0%); [n = 380] | |
| | | Bm14 Ab = 2.4% (1.3–4.5%); [n = 378] | |
| | Community survey (≥10 years) | MF = 0% (0–0.7%); [n = 528] | |
| | | ICT = 0% (0–0.7%); [n = 528] | |
| Rao et al. 2017[37] (Ambalangoda, Galle, Sri Lanka) | School survey (6–8 years) | MF = 0% (0–1.0%); [n = 379] | MIR = 1.17% (0.8–1.6%) [n = 3,993] |
| | | ICT = 0.3% (0–1.5%); [n = 379] | |
| | | Bm14 Ab = 2.3% (1.1–4.4%); [n = 353] | |
| | Community survey (≥10 years) | MF = 0.2% (0.3–1.0%); [n = 520] | |
| | | ICT = 1.0% (0.4–2.2%); [n = 520] | |
| Rao et al. 2017[37] (Unawatuna, Galle, Sri Lanka) | School survey (6–8 years) | MF = 0.3% (0–1.5%); [n = 359] | MIR = 1.23% (0.8–1.7%) [n = 4,002] |
| | | ICT = 1.1% (0.4–2.8%); [n = 359] | |
| | | Bm14 Ab = 4.2% (2.5–7.0%); [n = 333] | |
| | Community survey (≥10 years) | MF = 0.2% (0.0–1.0%); [n = 523] | |
| | | ICT = 1.5% (0.8–2.9%); [n = 523] | |
| Rao et al. 2017[37] (Matara, Sri Lanka) | School survey (6–8 years) | MF = 0% (0–1.0%); [n = 371] | MIR = 1.09% (0.7–1.5%) [n = 4,080] |
| | | ICT = 0% (0–1.0%); [n = 371] | |
| | | Bm14 Ab = 2.2% (1.1–4.2%); [n = 367] | |
| | Community survey (≥10 years) | MF = 0.2% (0.0–1.0%); [n = 525] | |
| | | ICT = 0.2% (0–1.0%); [n = 525] | |
| Rao et al. 2014[36] (Sri Lanka) | Community survey (≥10 years) | MF = 0–0.9% | MIR = 0–1.56% |
| | | ICT = 0–3.4% | |
| Rao et al. 2018[38] (Sri Lanka) | Community survey (10–70 years) | MF = 1.1% (0.5–2.5%) | MIR (2015) = 5.2% (4.2–6.3%) |
| | | ICT = 3.0% (1.8–4.9%) | MIR (2016) = 3.0% (2.3–3.8%) |
| Rao et al. 2016[33] (Sri Lanka) | Community survey (2–70 years) | MF = 0% (0.02–0.09%) | MIR = 0.36% (0.29%-0.45%) |

## Comparison with TAS results

18 studies reported alternative surveillance methods which were performed concurrently with, or subsequent to, a TAS which was passed successfully. The comparative results are illustrated in Table 6 which shows that alternative surveillance methods can identify evidence to support ongoing transmission in areas which passed TAS. For example, Sheel et al. report LF prevalence (using Filarial Test Strips) of 6.2% in a community survey in an area, which had recently passed TAS[14]. In American Samoa, Lau et al. (2014) found levels of Og4C3Ag to be 3.2% and Wb123 Ab to be 8.1% in an area which had recently passed TAS. Xenomonitoring surveys also appear to have utility in identifying hotspots, as in the case of Rao et al. (2018) who detected a MIR of 5.2% in an area which had recently passed TAS[38].

## Integration of surveillance with other disease programmes

The WHO recommend integrating post-MDA surveillance strategies with other ongoing surveillance activities[4]. Only three papers reported on efforts to integrate LF surveillance with other activities. A study from American Samoa tested stored bloods from a leptospirosis survey for LF[10]. Two studies from Togo integrated LF testing (using either MF or Og4C3 Ag) within routine malaria investigations either at the point of the diagnostic test being taken in the healthcare facility, or when the blood film was being analysed in the laboratory[40, 42].

**Table 6. Results of alternative surveillance conducted in settings which underwent concurrent TAS.**

| Country | Reference | Date passed TAS | Study date | Study type | Age | Sample size | Results (95% C.I.s if stated) |
|---|---|---|---|---|---|---|---|
| **American Samoa** | Lau et al. 2014 [11] | 2011 | 2010 | Community survey | ≥18 years | 807 participants | Og4cC3 Ag>32 units = 3.2% (0.6–4.7%); |
| | | | | | | | Wb123 Ab = 8.1% (6.3–10.2%) |
| | | | | | | | Bm14 Ab = 17.9% (15.3–20.7%) |
| | Schmaedick et al. 2014 [44] | 2011 | 2011 | Xenomonitoring survey | N/A | 15,215 mosquitoes | MIR rate = 0.28% (95% CI 0.20–0.39) |
| | Sheel et al. 2018 [14] | 2015 | 2016 | Community survey | ≥8 years | 2,507 participants | FTS = 6.2% (4.5–8.6%) |
| | | | | | | | MF = 22/86 +ve |
| | Won et al. 2018 [13] | 2011 | 2011 | Enhanced TAS[1] | 5–10 years | 1,134 participants | BinaxNOW = 0.2% |
| | | | | | | | Wb123 Ab = 1.0% |
| | | | | | | | Bm14 Ab = 6.8% |
| | | | | | | | Bm33 Ab = 12.0% |
| | | 2015 | 2015 | Enhanced TAS[1] | 5–10 years | 864 participants | BinaxNOW = 0.1% |
| | | | | | | | Wb123 Ab = 3.6% |
| | | | | | | | Bm14 Ab = 3.0% |
| | | | | | | | Bm33 Ab = 7.8% |
| **Bangladesh** | Irish et al. 2018 [45] | 2015 | 2016 | Xenomonitoring survey | N/A | 5,926 mosquitoes | MIR = 0% |
| **Egypt** | Moustafa 2014 [17] | 2012 | 2012 | Community survey | ≥18 years | 1,321 participants | BinaxNOW = 0% |
| | | | | | | | Bm14 Ab = 2.2% |
| **Madagascar** | Garchitorena et al. 2018 [25] | 2016 | 2016 | Community survey | ≥5 years | 545 participants | FTS = 15.78% (12.88–19.18%) |
| **Sri Lanka** | Rao et al. 2014a [36] | 2012–13 | 2012–13 | Community survey | ≥10 years | 7,156 participants | MF = 0–0.9% |
| | | | | | | | BinaxNOW = 0–3.4% |
| | Rao et al. 2014b [36] | 2012–13 | 2012–13 | Enhanced TAS[1] | 6–7 years | 17,000 participants | Bm14 Ab = 0–6.9% across school sites |
| | Rao et al. 2014c [36] | 2012–13 | 2012–13 | Xenomonitoring survey | N/A | 69,680 mosquitoes sampled | MIR = 0% - 1.56%. |
| | Rao et al. 2016 [33] | 2012–13 | 2014 | Xenomonitoring survey | N/A | 28,717 mosquitoes | MIR = 0.36% (0.29–0.45%). |
| | Rao et al. 2017c [37] | 2013 | 2015–17 | Community survey | ≥10 years | 3,123 participants (6 sites) | BinaxNOW = 0–1.5% |
| | | | | | | | MF = 0–0.2% (n.s.) |
| | Rao et al. 2017c [37] | 2013 | 2015–17 | School survey | 6–8 years | 2,227 participants (6 sites) | BinaxNOW = 0.0–1.1% |
| | | | | | | | MF = 0–0.3% |
| | | | | | | | Bm14 Ab = 0–4.2% |
| | Rao et al. 2017c [37] | 2013 | 2015–16 | Xenomonitoring survey | N/A | 24,061 mosquitoes (6 sites) | MIR = 0.23% (Peliyagoda) - 1.23% (Unawatuna) |
| | Rao et al. 2018d [38] | 2013 | 2015–16 | Xenomonitoring survey | N/A | 2015: 4,000 mosquitoes | 2015: MIR = 5.2% (4.2–6.3%). |
| | | | | | | 2016: 3,750 mosquitoes | 2016: MIR = 3.0% (2.3–3.8%). |
| | Rao et al. 2018a [38] | 2013 | 2015 | School survey | 6–7 years | 401 participants | BinaxNOW = 1.2% (0.5–2.8%) |
| | | | | | | | MF = 0.2% (0.0–1.4%) |
| | | | | | | | Bm14 Ab = 5.7% (3.7–8.4%) |
| | Rao et al. 2018b [38] | 2013 | 2015 | Community survey | 10–70 years | 528 participants | BinaxNOW = 3.0% (1.8–4.9%) |
| | | | | | | | MF = 1.1% (0.5–2.5%) |
| | Rao et al. 2018c [38] | 2013 | 2015 | Community survey | ≥2 years | 16,927 participants | MF = 0.6% (0.47–0.71%) |

(*Continued*)

**Table 6.** (Continued)

| Country | Reference | Date passed TAS | Study date | Study type | Age | Sample size | Results (95% C.I.s if stated) |
|---------|-----------|-----------------|------------|------------|-----|-------------|-------------------------------|
| **Togo** | Dorkenoo et al. 2018B [52] | 2015 | 2015 | Xenomonitoring survey | N/A | 10,872 mosquitoes | MIR = 0%. |

[1] Standard TAS with the addition of antibody testing

## Discussion

This review provides a timely collation of important information on alternative surveillance strategies for low prevalence and/or post-validation settings that will be useful to national programmes over the next decade as they seek to reduce LF incidence and meet the challenges of the NTD Roadmap 2030 [54]. However, the significant heterogeneity found in the study designs, population sampled, use of diagnostic tests and reporting metrics, highlights the need for more systematic methods and new WHO guidelines to be developed to supplement TAS.

This review has identified that the sensitivity of LF surveillance in selected low prevalence populations can be increased by changes to the diagnostic test and/or study population. TAS is an important programmatic tool to guide decisions on when to stop MDA but several studies report that it lacks sensitivity when used in low prevalence settings, such as a post-validation context[13, 17, 23, 38], and may not accurately describe the spatial distribution of LF at community-level[14]. This is important because evidence from countries that have recently eliminated LF indicates an increased risk of disease recrudescence, with ongoing hotspots of infection documented recently in both American Samoa and Sri Lanka[5, 12, 38]. The lag time between infection with LF and onset of symptoms may be 10 years or more, demonstrating the critical importance of maintaining surveillance programmes following elimination[41, 51]

### Alternative diagnostic tests

The studies included in this review indicate that there may be benefit in moving from the conventional rapid antigen tests to antibody tests as they increase the proportion of positive results and, hence, the likelihood that residual hotspots will be detected. However, antibody tests are a measure of the host response to infection which can persist for some time after all antigenic material from the original infection has been eliminated. This means that antibody tests are associated with an increased false-positive rate and the detection of more historical cases, meaning there would be financial and logistical implications to switching to widespread antibody testing[13].

Antibody tests could be added to TAS without significant changes to study design[13]. Reported results suggest that testing Wb123 antibody (Ab) may have particular utility since it is thought to both become positive relatively soon after infection and decay faster following clearance, compared to Bm14 Ab[27, 55]. It also has been found to be significantly associated with molecular xenomonitoring results, suggesting it could act as an indicator of ongoing transmission[13, 55]. Urine ELISA may have greater acceptability than blood testing but requires further validation in LF-endemic regions[16, 35]. However, the current increased costs of antibody and ELISA tests may limit their widespread uptake and further research is needed to characterise the spatial distribution of antibody signals[13].

All methods of human surveillance are affected by the persistence of the marker (antibody or antigen) in circulation. This is of variable duration for different test types, meaning that their results are not directly comparable. It also means that results are not truly indicative of

current infectivity and will therefore include cases of historical disease. By contrast, mosquito surveillance gives a snapshot indication of current infection, and could serve as a useful adjunct to human surveillance methods[27, 47]. However, mosquito surveillance requires entomology and laboratory capacity, which are both costly and time-consuming, meaning that it is typically only used in very defined areas, rather than for population-wide surveillance[27, 45].

## Alternative approaches to sampling

Studies reported that LF tests typically report higher prevalence of infection in adults than in children[14, 31] and it is thought that adults (particularly adult men) may represent the majority of the reservoir of infection for LF[37]. As prevalence reduces it may therefore also be appropriate to target surveillance to focus on these high-risk populations. Methods that have been suggested include adopting a 'test and treat' approach for adult males, which could focus on settings in which they may be more likely to congregate, such as marketplaces[37].

Post-validation surveillance in Togo found positive cases in low-risk areas, highlighting the importance of developing surveillance systems with nationwide coverage[40, 42]. Areas with high levels of migration from endemic countries (e.g. border areas) may also require additional monitoring[24, 56]. Other recommended sampling methods include community-based methods targeting adults and children, school-based surveys with a wider age range and snowball sampling of positive cases[14].

## Future research needs

In order to support countries to develop appropriate surveillance in low prevalence or post-validation settings, further research will be required to inform choices regarding the selection of diagnostic tests and appropriate sampling strategies. This will include work to determine the diagnostic performance and cost-effectiveness of novel tests in a range of different epidemiological settings and the identification of suitable threshold values for new LF diagnostic tests in humans[13]. Further research is also required to determine appropriate sample size and infection cut-off thresholds for surveillance in different mosquito species[18, 26, 44, 52].

There is a need to better understand the spatial and temporal dynamics of LF hotspots and their drivers, which will require more longitudinal studies to help inform future control and surveillance activities[12, 23]. Emerging evidence suggests that LF hotspots may be highly focal, increasing the likelihood that cluster-based methods will lack sensitivity to detect them [10, 12, 23, 57]. The risk of recrudescence of infection will depend on a range of factors including population density, baseline endemicity, uptake of MDA and concurrent vector control interventions. It may be appropriate to stratify the intensity of population-level surveillance based on assessment of these factors[58]. This must be supported by the development of data systems capable of continuously collecting, analysing and interpreting data in order to rapidly inform service planning and policy[6].

Further, there is a particular need to increase the evidence base in the African and South Asian Regions, which currently have the majority of ongoing transmission[1]. The evidence base supporting integration of surveillance activities with other health system processes must also be strengthened. Examples may include blood donation systems, surveillance for other co-endemic NTDs (e.g. onchocerciasis) or malaria and routine household surveys[24, 40, 41, 48]. Finally, post-validation surveillance programmes will require clear guidance on how to respond to the identification of new cases. Such interventions may include watchful waiting, vector control, resumption of MDA, treatment of cases only, or a combination of methods [36].

## Limitations

It was not possible to conduct a meta-analysis of surveillance results which was largely due to the variation in study methods, but also because of the variation in the infectivity of different mosquito vectors and the influence of different environmental factors that are difficult to control for.

Regarding the study exclusion criteria, the decision to limit the analysis to the English language led to the exclusion of a small number of papers published in Spanish or Chinese, but we consider it unlikely that these results would have significantly changed the main outcomes of this review. The decision to limit the review to papers published after 2000 also excluded a small number of papers but it was considered that the results of more historical studies were likely to have limited transferability to current LF programmes. Finally, our search for unpublished data was limited. It is likely that some studies examining surveillance methods are conducted as part of routine LF programmatic activities and, hence, not published. If collected, such data could strengthen the evidence base in this area.

## Conclusions

This is the first review to systematically investigate the evidence supporting alternative (non-TAS) approaches to LF surveillance in low prevalence and post-validation settings. The results demonstrate a need for a more standardised approach to LF surveillance in low prevalence and post-validation settings. Surveillance methods with greater sensitivity and more targeted sampling strategies to better detect residual hotspots than the current TAS methodology will be required. However, further research on the diagnostic performance and cost-effectiveness of new diagnostic tests, and how these can be integrated within routine health system activity, is needed to inform policy decisions over the next decade.

## Supporting information

**S1 File. PRISMA checklist.**
(DOC)

**S2 File. Search strategy.**
(DOCX)

**S1 Table. Risk of bias assessment for human and mosquito studies.**
(DOCX)

**S2 Table. Human surveillance study results.**
(DOCX)

**S3 Table. Mosquito surveillance study results.**
(DOCX)

## Author Contributions

**Conceptualization:** Louise A. Kelly-Hope.

**Formal analysis:** Nicholas Riches, Xavier Badia-Rius.

**Investigation:** Nicholas Riches, Xavier Badia-Rius.

**Methodology:** Nicholas Riches, Louise A. Kelly-Hope.

**Project administration:** Nicholas Riches.

**Supervision:** Louise A. Kelly-Hope.

**Visualization:** Nicholas Riches, Xavier Badia-Rius.

**Writing – original draft:** Nicholas Riches.

**Writing – review & editing:** Nicholas Riches, Xavier Badia-Rius, Themba Mzilahowa, Louise A. Kelly-Hope.

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
