## [Decision Letter · Decision Letter 0]

10 Oct 2019

Dear Dr Riches:

Thank you very much for submitting your manuscript "A systematic review of alternative approaches to lymphatic filariasis post-elimination surveillance" (#PNTD-D-19-01432) for review by PLOS Neglected Tropical Diseases. Your manuscript was fully evaluated at the editorial level and by independent peer reviewers. The reviewers appreciated the attention to an important problem, but raised some substantial concerns about the manuscript as it currently stands. These issues must be addressed before we would be willing to consider a revised version of your study. We cannot, of course, promise publication at that time.

We therefore ask you to modify the manuscript according to the review recommendations before we can consider your manuscript for acceptance. Your revisions should address the specific points made by each reviewer. 

When you are ready to resubmit, please be prepared to upload the following:

(1) A letter containing a detailed list of your responses to the review comments and a description of the changes you have made in the manuscript.

(2) Two versions of the manuscript: one with either highlights or tracked changes denoting where the text has been changed (uploaded as a "Revised Article with Changes Highlighted" file); the other a clean version (uploaded as the article file).

(3) If available, a striking still image (a new image if one is available or an existing one from within your manuscript). If your manuscript is accepted for publication, this image may be featured on our website. Images should ideally be high resolution, eye-catching, single panel images; where one is available, please use 'add file' at the time of resubmission and select 'striking image' as the file type. 

Please provide a short caption, including credits, uploaded as a separate "Other" file. If your image is from someone other than yourself, please ensure that the artist has read and agreed to the terms and conditions of the Creative Commons Attribution License at http://journals.plos.org/plosntds/s/content-license (NOTE: we cannot publish copyrighted images). 

(4) If applicable, we encourage you to add a list of accession numbers/ID numbers for genes and proteins mentioned in the text (these should be listed as a paragraph at the end of the manuscript). You can supply accession numbers for any database, so long as the database is publicly accessible and stable. Examples include LocusLink and SwissProt.

(5) To enhance the reproducibility of your results, we recommend that you deposit your laboratory protocols in protocols.io, where a protocol can be assigned its own identifier (DOI) such that it can be cited independently in the future. For instructions see http://journals.plos.org/plosntds/s/submission-guidelines#loc-methods

While revising your submission, please upload your figure files to the Preflight Analysis and Conversion Engine (PACE) digital diagnostic tool, https://pacev2.apexcovantage.com/ PACE helps ensure that figures meet PLOS requirements. To use PACE, you must first register as a user. Then, login and navigate to the UPLOAD tab, where you will find detailed instructions on how to use the tool. If you encounter any issues or have any questions when using PACE, please email us at figures@plos.org.

We hope to receive your revised manuscript by Dec 09 2019 11:59PM. If you anticipate any delay in its return, we ask that you let us know the expected resubmission date by replying to this email.

To submit a revision, go to https://www.editorialmanager.com/pntd/ and log in as an Author. You will see a menu item call Submission Needing Revision. You will find your submission record there. 

Sincerely,

Patrick J. Lammie, Ph.D.

Associate Editor

Jennifer Keiser

Deputy Editor

In preparing this manuscript for re-submission, it is important to address the concerns raised by all three reviewers.

Reviewer's Responses to Questions

**Key Review Criteria Required for Acceptance?**

**Methods**

-Are the objectives of the study clearly articulated with a clear testable hypothesis stated?

-Is the study design appropriate to address the stated objectives?

-Is the population clearly described and appropriate for the hypothesis being tested?

-Is the sample size sufficient to ensure adequate power to address the hypothesis being tested?

-Were correct statistical analysis used to support conclusions?

-Are there concerns about ethical or regulatory requirements being met?

Reviewer #1: The objective is to conduct a systematic review. This is not a hypothesis-based study although some hypotheses are implicit in the document (do adults have higher prevalence than children, for example). The study design is appropriate. Sample size depends on the studies reviewed and there are no ethical concerns. 

The aim is stated to be to make recommendations to programme managers and highlight areas regarding further research. I don't think any clear recommendations (particularly to programme managers) come out of the review at present. I believe there may be useful summary observations that could be drawn out. 

The paper contains a large amount of important information, and it is very useful to have this information collected together, but there are three main areas where improvement is needed to make this a more coherent and valuable contribution. 

1. Terminology and organization. There are two relevant time periods for the studies reviewed. Post-MDA - after a country or area believes they can stop MDA, and post validation - after further time has elapsed and surveys have been conducted. The authors use the word 'elimination' usually to describe the stage of the process known as 'validation'. This is confusing and needs to be changed throughout. Since countries and areas are stopping at different times, it would help to show when each site reached these two milestones - i.e. present a table showing the country or area timelines of stopping MDA and validation (rather than just calendar dates of studies, as in Table 1) and then present results by the same timelines. Perhaps this might illuminate some insights about when it is best to do surveys (some are right after last MDA) or how things change over time since MDA or validation. 

2. Presentation and assessment of data. The authors state that they have not done and cannot do formal risk of bias assessment or formal metaanalysis. But there are tools available to do risk of bias assessment of such studies. An example is the Crowe Critical Appraisal tool or the Gate Frame. Studies can be ranked by sample size (small, med, large), method of sampling of sites (simple random, cluster, convenience, etc), number of sites (one village versus many), sampling of participants or mosquitoes (representativeness), methodology used for detecting infection, etc. At the very least, some key features of the studies and their quality should be reported in the tables rather than just summing up how many were random sampling (which ones?) and how many weren't. I am not sure if I am convinced that formal metanalysis could not have been done in some cases, but even if not, please at least borrow from the metananalysis field for data presentation, and use forest plots to present the prevalence estimates rather than solely tables. Trying to glean useful summary information from long tables such as Table 3 and 4, or 7 is very difficult for the reader. Showing forest plots graphically (ideally with CIs even if unadjusted), organised by strata such as time since MDA, age group or region/vector would be much more useful. 

3. Discussion in context. Without some more effort to appraise the study quality or synthesize the results, even broadly, the true additional value of the paper is not clear. In the Discussion, a paper is supposed to show how the results from this paper add to the available literature or concepts. The authors revert back to citing individual studies to support their points rather than their own Results, which doesn't move us forward much. What can we conclude that is not already known? It leaves the impression that the Discussion could mostly have been written as a general review document before the systematic review was done. I think the hard work put into this deserves a more reflective, concise and nuanced Discussion than that.

Reviewer #2: This manuscript by Riches et al. is helpful in providing a list of published studies of interest to those who are tasked with determining which surveillance strategies would be most useful for defining the effectiveness of programs aimed at eliminating lymphatic filariasis. It does, however, lack so much important detail and conceptual context that it is unable to break much new ground or even to guide those not already very familiar with LF and its elimination in the most fruitful directions.

Principal problems with the manuscript:

1) Critical terms are poorly or not-at-all defined; these include TAS, enhanced TAS, elimination, post-elimination, post-MDA, elimination as a public health problem, elimination of transmission, transmission itself, validation, verification, and others

a. each of these terms is important, has a precise definition and relates in a very distinct way to the surveillance targets and challenges for LF

b. any review of approaches to surveillance for LF must include a clear identification of the goals being targeted at each stage of the program, and those goals demand clear definitions for these terms

2) The various ‘alternatives’ to TAS presented are a collection of studies with very different ‘context.’ The authors recognize this challenge and do not want to introduce bias, but just describing the studies and not adding some weighting of their importance based on context doesn’t much help in determining the most meaningful conclusions from the data 

a. ‘Post-MDA’ assessment can imply any of a number of different time points, and the same can be said of ‘post-elimination’ assessments

b. sampling strategy, age-group sampled and diagnostic tools are variables that the authors do wrestle with, but the lack of uniformity in the studies makes many potentially important comparisons nearly impossible to make (especially those involving TAS comparisons).

3) In discussing the various diagnostic tests, higher positivity rates are considered to reflect greater sensitivity of the diagnostic. What is not known, however, is how diminished specificity affects the interpretation of these findings. Much is still unknown about the specificity, sensitivity and kinetics of antibody tests in particular.

Reviewer #3: (No Response)

**Results**

-Does the analysis presented match the analysis plan?

-Are the results clearly and completely presented?

-Are the figures (Tables, Images) of sufficient quality for clarity?

Reviewer #1: Please see comment above regarding better presentation of study data as forest plots (available in RevMAN, STATA etc) to make the results clearer. Even if using tables, consider organizing or stratifying results by time post MDA and post validation, by type of sampling/quality of study, age group or other factor, rather than just by country and calendar year. Include at least minimal quality scores or quality features in descriptive tables or stratify by study type/quality in plots if possible. 

Please be clearer about the term 'diagnostic yield'. e.g. in title of Table 4, line 207 and elsewhere. It is not a standard term. Perhaps just use 'estimated prevalence'? The authors need to clarify that tests for Mf, Ag (ICT, FTS, Og4C3) and Ab (various types) are testing for different things. The sensitivity or diagnostic accuracy of the diagnostic tests is not really being compared here.

Table 4 would be better to me if the different markers were separated out and presented by the different age groups. Perhaps simplify to just adults and children only, or 3 age groups only. Table 5 is even more obscure since it has both age and gender for one study. If prevalence by both age and gender are important and confounded, please separate out the studies that have both age and gender from just one or the other. Arrangement of the studies is not optimally logical for data synthesis or putting similar outcomes together.

Reviewer #2: see above

Reviewer #3: (No Response)

**Conclusions**

-Are the conclusions supported by the data presented?

-Are the limitations of analysis clearly described?

-Do the authors discuss how these data can be helpful to advance our understanding of the topic under study?

-Is public health relevance addressed?

Reviewer #1: The paper's conclusions do not seem very well justified by the results, and are vague. They state need for more research and more standardized approaches, as well as for methods with greater diagnostic sensitivity and alternative sampling strategies, but don't give any suggestion about what these approaches should be. I am not convinced by the conclusion about diagnostic sensitivity. It is not the diagnostic methods, but when, whom and how you sample, what is measured, and what the thresholds are, that are more important. It might be good to have novel tests, but to solve what problem exactly? I think there are better and bolder conclusions that could be reached from the study.

Reviewer #2: (No Response)

Reviewer #3: (No Response)

**Editorial and Data Presentation Modifications?**

Reviewer #1: I understand the reasonable limitations and the need for a date when the search had to stop, but there still seem to be some missing studies. Maybe there were reasons for exclusion. 

for example

Post MDA studies in Lihir, island PNG (Mitja et al PLOS NTD 2011)

PNG: You have included the Tisch studies in 1990s which seem outside the time frame of publication starting 2000, but what about the later work by Riemer et al in the same area ? 

Hapairai et al 2015 in Samoa (xenomoitoring) Parasites and Vectors 2015 

French Polynesia: the Maupiti work, Esterre 2001 and 2005 (in English), maybe other areas in Fr Poly that stopped MDA?

Cook Islands: Ave et al 2018 Trop Med Health

Vanuatu: Taleo et al 2017, Trop Med Health

Gass et al 2012 (many countries)

An important paper not cited regarding xenomonitoring thresholds is by Pedersen et al 2009 Trends Parasitol

https://www.sciencedirect.com/science/article/pii/S1471492209001160?via%3Dihub

Minor error in Table 3 last line under Joseph et al: Vanuatu results are put under Tonga

Reviewer #2: see above

Reviewer #3: (No Response)

**Summary and General Comments**

Reviewer #1: Generally this is a comprehensive study demonstrating thorough and apparently careful work and extraction of useful information from a large number of diverse studies that are useful to see together. To maximise its value the paper needs to present the data more clearly and logically to enable 'side by side' comparison of studies with similar outcomes or characteristics. It needs better organization and presentation in relation to MDA and validation timelines, some kind of critical appraisal of the studies, and effort to synthesize at least some combined broad interpretations which arise from the data in Tables or Figures and which are more available for the readers to assess themselves. The Discussion needs to specifically relate the findings in this paper to the overall status of LF elimination programmes, strategies and thresholds, rather than quoting individual studies again, and show how this study moves us forward with more specific recommendations about sampling strategies, methods or other improvements.

Reviewer #2: While the collection of studies reviewed in this manuscript will be helpful to those aiming to draw inferences and testable hypotheses from the information already available, it is unfortunate that these authors did not do the further analytic work to define many of these inferences and testable hypotheses themselves. It is clear from their other manuscripts that this group does have the technical and conceptual background to do such analytic work.

Reviewer #3: The authors have conducted an interesting literature review, dissecting various methods of LF surveillance in research studies after 2000. 

It’s useful to see all that has been done and the results collated in one paper. However, other than categorizing and presenting results of any published non-TAS survey since 2000, it is not clear what question, if any is being asked. Can the authors articulate what would have the formal meta-analysis measured?

The review would be stronger if hypotheses could be tested and forest plots generated. 

When referring to elimination of LF, please indicate elimination as a public health problem

Line 45: the review assesses the ‘results’ from 42 studies

Line 46-47: There is no standardized approach to testing ‘other than TAS’

Line 189. How does the variation in the sampling methods allow direct comparison of prevalence?

Line 204. Define noticeably. Was higher prevalence noticeable in all locations?

Line 208 second part of the sentence is a discussion point not a result.

Table 6 could the authors add the primary vector? 

Line 237-238 define appear more sensitive. Should this interpretation be moved to the discussion?

Line 253-263 Indicates the main purpose of the paper was to conclude TAS lacks sensitivity. Where was this stated as an objective and how was this measured? Where was the evidence on size of evaluation units? 

If revised, the statement starting in line 261 to 263 could be a conclusion of the review and would be ideal to put as the start of the discussion. It would be important for the authors to define sensitivity (or how they are using it) in the methods. More positive test results? 

Line 284-286 – are the authors claiming that the results from the review suggest test-and treat strategy of males? Please clarify on which data this recommendation is based or remove.

Line 294 Future research needs – it is not clear whether these were derived from the studies reviewed. It would strengthen the paper if the authors noted / included in the results the various research needs identified in the included studies. As it is simply listed in the discussion it seems as the authors’ opinions of the needs. 

Line 304 – do the authors mean recrudescence of infection or disease? 

Conclusions

Line 334 – 335 the authors join two separate conclusions. Please revise to tease out the 2 points being made.

PLOS authors have the option to publish the peer review history of their article (what does this mean?). If published, this will include your full peer review and any attached files.

Reviewer #1: No

Reviewer #2: No

Reviewer #3: No

---

## [Decision Letter · Decision Letter 1]

23 Feb 2020

Dear Dr Riches,

Thank you very much for submitting your manuscript "A systematic review of alternative surveillance approaches for lymphatic filariasis in low prevalence settings: implications for post-validation" for consideration at PLOS Neglected Tropical Diseases. As with all papers reviewed by the journal, your manuscript was reviewed by members of the editorial board and by several independent reviewers. The reviewers appreciated the attention to an important topic. Based on the reviews, we are likely to accept this manuscript for publication, providing that you modify the manuscript according to the review recommendations. 

Sincerely,

Patrick J. Lammie, Ph.D.

Associate Editor

Jennifer Keiser

Deputy Editor

Reviewer's Responses to Questions

**Key Review Criteria Required for Acceptance?**

**Methods**

-Are the objectives of the study clearly articulated with a clear testable hypothesis stated?

-Is the study design appropriate to address the stated objectives?

-Is the population clearly described and appropriate for the hypothesis being tested?

-Is the sample size sufficient to ensure adequate power to address the hypothesis being tested?

-Were correct statistical analysis used to support conclusions?

-Are there concerns about ethical or regulatory requirements being met?

Reviewer #1: The authors have been very responsive and addressed the main deficiences with the previous version. The presentaton is greatly improved and the inclusion of risk of bias/quality assessment is welcomed.

Post validation is now clear, but the definition of 'post MDA' is still not clear and a bit inconsistent. For example, in Table 2 you have included studies from countries that are not truly 'post-MDA'. For example Samoa in 2008 was after several MDAs, but the surveys by Joseph et al were done before (or around the same time as) the 2008 MDA, which was followed by MDAs in 2011 and later. Other countries like Nigeria, PNG, Haiti, Fr Poly and others are not 'post MDA'. American Samoa has restarted in 2018. I don't think you need to change which studies are included, but need to come up with a better term than 'post MDA' to describe them and why you included them. Solomons was not post -validaton becasue it did not need MDA. 

Also please define 'enhanced TAS'.

**Results**

-Does the analysis presented match the analysis plan?

-Are the results clearly and completely presented?

-Are the figures (Tables, Images) of sufficient quality for clarity?

Reviewer #1: Much clearer overall, but Table 6 is the exception. It's very obscure and hard to understand which surveys go together. 

Please put the surveys side by side in Table 6. 

Table S4 helps a bit - but requires effort for the reader to find. Maybe it should be in main paper if you are gonig to draw conclusions from it (as in abstract you do - but not in text). 

The description of results and their interpretation from Table S4 and Table 6 are very skimpy . A table 7 is cited but not present.

You have made quite a bold statement in the abstract that MX is better at determining ongoing transmission. Where is the evidence for that from these tables? MX could also just be detecting MF in non competent vectors - the word 'mosquito transmission' in line 308 should be 'mosquito infection'. What is a 'successful' TAS? (line 259). 

I am surprised you didn't include the paper by Lau et al 2016 in table S4 that directly compared the serology and MX results from American Samoa in roughly the same time frame as the MX and TAS in 2011 was done . It is is cited in the Won et al paper (13) in lines 300-303, but the actual comparison is in Lau et al 2016 PLOS NTD . If you are going to rely on this result so much for Wb123 please cite the original paper. 

I like the trend lines in Table 4, but they sometimes don't seem to relate exactly to the data in the precending columns. Why no trend in the first row? For Lau et al 2014, there are 6 points on the line, but no data for the first two in the table. Same for Rao et al and Sheel et al missing 0-10 yrs.

Figure 3 with gender specific prevalence is great, but there seem to be some studies missing that reported gender . Lau et al 2014, 2017, maybe others.

**Conclusions**

-Are the conclusions supported by the data presented?

-Are the limitations of analysis clearly described?

-Do the authors discuss how these data can be helpful to advance our understanding of the topic under study?

-Is public health relevance addressed?

Reviewer #1: Conclusion is much better and is anchored in the results/discussion. However still not very clear throughout on the definiton of diagnostic sensitivity and its difference from surveillance sensitivity. The Ag/Ab tests are detecting different things so it is not just differences in persistence or incubation period, as implied in Discusion.

Need to be very clear on sensitivity of surveillance methods versus sensitvity of tests. i.e. perhaps remove the word diagnostic line 364. Please check lines 135 and 136 as they do not seem correct either. What is the true positive here?

I am not clear on the definition of 'clinical effectiveness' - used in conclusion but also elsewhere. Please explain what you mean by this.

**Editorial and Data Presentation Modifications?**

Reviewer #1: Please check use of 'paper/article', 'study' and 'study design' . 

There are 44 articles and 83 studies. Lines 159 - 162 are confusing. Papers reported data from 22 countries?; 21 STUDIES came from WPRO etc,. 

45 articles are mentioned line 168 when I think you mean 45 studies. 

 Legend to Fig 2 - I think you mean '60 distinct studies' not 'study designs'

There may be others.

Table S1 - Lau et al 2014 was actually simple random sample of households (not non-random). This will change the quality score. 

Lau et al 2017 - What is definition of 'children'? line [3] was adult workers, only a few of whom were 16 -17 yrs old. 

Please also check Joseph et al 2011 (nos 22 and 23). One of those was children only. Those are just the ones I know about.

Table 2 - please review 'Context' and consider a better classification. e.g. Just finishing MDA (doing a survey of all ages to check if can stop e.g. Joseph Samoa, Allen Vanuatu, Coutts Am Samoa); in a lull between MDAs and waiting for validation; doing a pre-TAS, never did MDA but checking, etc.

Table S2. same comments on Context column.You still have 'post-elimination' in the context here - do you mean post-validation?

Gass et al ref is 2012 not 2011 (study done 2011). 

Mitja et al [30] is PNG not Tanzania. 

Line 204 suggest change for clarity to 

"Compared to Binax Now or Alere ICT (most commonly used tests at the time of most of these surveys) as the index test, Table 3 shows ..."

Line 308 suggest:

"Post validation surveillance in Togo found positive cases in low- risk areas" to cut repetition of 'found'

Fig 3 can you match the colours of % legends to the bars? It's unecessarily confusing right now. 

Table S4 - define MIR

**Summary and General Comments**

Reviewer #1: Overall the aims are much clearer, the data presentation is greatly improved (with a few exceptions already mentioned earlier) and the paper is much more comprehensible and useful.

PLOS authors have the option to publish the peer review history of their article (what does this mean?). If published, this will include your full peer review and any attached files.

Reviewer #1: No
---

## [Editor Report · Decision Letter 2]

13 Apr 2020

Dear Dr Riches,

We are pleased to inform you that your manuscript 'A systematic review of alternative surveillance approaches for lymphatic filariasis in low prevalence settings: implications for post-validation' has been provisionally accepted for publication in PLOS Neglected Tropical Diseases.

Best regards,

Patrick J. Lammie, Ph.D.

Associate Editor

Jennifer Keiser

Deputy Editor

I appreciate the careful attention that the authors have focused on an important topic as well as their diligence in responding to reviewer suggestions. Nonetheless, I think that the manuscript would still benefit from a careful proof reading.

As examples:

Line 33: There is a loose “h” at the end of the line.

Table 4: The cells with the trend lines do not match up with the rows in all instances.

Line 256: The sentence ends with a double period.

Lines 257-258: The phrase “but interestingly not in other studies in Sri Lanka” requires further explanation.

---

## [Editor Report · Acceptance letter]

23 Apr 2020

Dear Dr Riches,

We are delighted to inform you that your manuscript, "A systematic review of alternative surveillance approaches for lymphatic filariasis in low prevalence settings: Implications for post-validation settings," has been formally accepted for publication in PLOS Neglected Tropical Diseases.

Best regards,

Serap Aksoy

Editor-in-Chief

Shaden Kamhawi

Editor-in-Chief
